# Disentangling bacterial invasiveness from lethality in an experimental host-pathogen system

Tommaso Biancalani & Jeff Gore*  iD

## Abstract

Quantifying virulence remains a central problem in human health, pest control, disease ecology, and evolutionary biology. Bacterial virulence is typically quantified by the *LT50* (i.e., the time taken to kill 50% of infected hosts); however, such an indicator cannot account for the full complexity of the infection process, such as distinguishing between the pathogen's ability to colonize versus kill the hosts. Indeed, the pathogen needs to breach the primary defenses in order to colonize, find a suitable environment to replicate, and finally express the virulence factors that cause disease. Here, we show that two virulence attributes, namely pathogen lethality and invasiveness, can be disentangled from the survival curves of a laboratory population of *Caenorhabditis elegans* nematodes exposed to three bacterial pathogens: *Pseudomonas aeruginosa*, *Serratia marcescens,* and *Salmonella enterica*. We first show that the host population eventually experiences a constant mortality rate, which quantifies the lethality of the pathogen. We then show that the time necessary to reach this constant mortality rate regime depends on the pathogen growth rate and colonization rate, and thus determines the pathogen invasiveness. Our framework reveals that *Serratia marcescens* is particularly good at the initial colonization of the host, whereas *Salmonella enterica* is a poor colonizer yet just as lethal once established. *Pseudomonas aeruginosa*, on the other hand, is both a good colonizer and highly lethal after becoming established. The ability to quantitatively characterize the ability of different pathogens to perform each of these steps has implications for treatment and prevention of disease and for the evolution and ecology of pathogens.

**Keywords** *C. elegans*; host–pathogen; microbial ecology; population dynamics; virulence

**Subject Categories** Microbiology, Virology & Host Pathogen Interaction; Quantitative Biology & Dynamical Systems

**Mol Syst Biol. (2019) 15: e8707**

## Introduction

Quantifying virulence is challenging because the mortality induced by a pathogen is determined by a complex series of interactions between the pathogen and the host (Méthot & Alizon, 2014). The virulence of a pathogen will depend upon the level of *invasiveness*, governed by the pathogen's ability to colonize and grow within the host, as well as the level of *lethality*, governed by the mortality that is induced following colonization due to factors such as toxicity (Casadevall & Pirofski, 1999). Moreover, both the invasiveness and lethality of a pathogen will depend upon host characteristics such as age, immune system, and microbiome (Casadevall & Pirofski, 2001). Historically, it has been proposed that virulence attributes are controlled by evolutionary trade-offs (Anderson & May, 1982; Alizon *et al*, 2009), and theoretical studies have demonstrated that they constrain the evolutionary (Roy & Kirchner, 2000) and ecological (Sansonetti, 2011) fate of the host–pathogen system.

Despite these results, there is still no consensus on how to disentangle the various pathogen attributes from the survival curves of a host–pathogen system. In the majority of experimental studies, survival curves are merely boiled down to the phenomenological indicator *LT50*, which denotes the median lethal time (e.g., in pest control (Abrol, 2013) and toxicology (Comprehensive Toxicology—3rd Edition)). The usage of *LT50* is often justified on grounds of simplicity, despite the fact that this indicator suffers from being highly sensitive to experimental conditions (as shown later). Most importantly, *LT50* does not describe a specific characteristic of the pathogen but rather provides a rough account of all factors that cause virulence. Hence, it is not possible to use *LT50* to disentangle whether the hosts are dying because of a highly invasive pathogen or a highly lethal one. The question can also be posed conversely: What pathogen attributes need to be known to fully determine the survival curves of the hosts?

Here, we use an experimentally tractable host–pathogen model system to disentangle how pathogen invasiveness and lethality lead to pathogen virulence. We study the dynamics of a laboratory population of hosts (the nematode *C. elegans*) exposed to three bacterial human pathogens that also cause mortality in *C. elegans*: *P. aeruginosa*, *S. marcescens,* and *S. enterica*. This experimental system, especially with the pathogen *P. aeruginosa*, has been used to investigate molecular mechanisms of virulence (Mahajan-Miklos *et al*, 1999; Tan *et al*, 1999), animal immunity (Kim *et al*, 2002), and mechanisms for pathogen aversion (Zhang *et al*, 2005). Quantitative analysis of survival curves of worms exposed to all three pathogens revealed that the worms eventually experienced a host–pathogen-specific per capita mortality rate. This pathogen-specific mortality rate was independent

Physics of Living Systems, Department of Physics, Massachusetts Institute of Technology, Cambridge, MA, USA
*Corresponding author. Tel: +1 617-715-4251; E-mail: gore@mit.edu

of pathogen exposure, indicating that it reflects the intrinsic lethality of the pathogen against this host. A theoretical model incorporating host colonization and pathogen growth predicts that the constant host mortality rate emerges from pathogen load saturating within the host, and this prediction is confirmed experimentally. The time necessary to reach this exponential phase where the host experiences a constant mortality rate reflects the pathogen invasiveness, due to the pathogen colonization rate and growth rate within the host. Our integrated experiments and modeling approach therefore allow us to disentangle the invasiveness from the lethality and to see how each quantitatively depends upon the pathogen colonization rate, the growth rate within the host, and the pathogen lethality.

## Results

### Experiments show that survival curves display an exponential phase

Our initial aim was to analyze how the *C. elegans* host survival curves are affected by exposing the hosts to the same pathogen at different pathogen densities. On agar plates with rich media, we spread a lawn of *P. aeruginosa* and incubated for either 4 h (low), 24 h (mid), or 48 h (high), so that the pathogen density can reach different densities (e.g., high = high pathogen density). On each agar plate, we then added a population of approximately fifty *C. elegans* adult nematodes, which are same age, reproductive sterile, and initially germ-free. The nematodes feed on the pathogens, which colonize the worm gut and disrupt the epithelium provoking the death of the host. Using standard worm picking protocols, we monitor the fraction of worms surviving over time (Kirienko *et al*, 2014; see Materials and Methods).

As expected from previous experimental results (Clark & Hodgkin, 2014), the worms die due to bacterial infection over the course of a few days, whereas in the absence of the pathogen, the worms would live for a few weeks. Consistent with the expectation that higher bacterial densities will be more virulent, we find that the lethal time for 50% of worms to die (*LT50*) is approximately 95 h for worms fed at low bacterial density (that is, pre-incubated for 4 h) and 55 h for worms fed at high bacterial density (pre-incubated for 48 h; Fig 1A). The measured *LT50* therefore depends not only on the particular pathogen and host being studied, but also on the details of the experimental protocol, in this case the pre-incubation time of the pathogen.

It would be ideal if there were some features in the survival curves that were independent of the pathogen lawn density, as this would indicate an attribute that was intrinsic to the pathogen and its host. Encouragingly, the survival curves plotted on a semi-log scale show a linear regime, indicating that over longer times the worms are dying at a constant (per capita) rate (Fig 1A). In this regime, the fraction of worms surviving decays exponentially; thus, we refer to this regime as the exponential phase. The slopes of the survival curve lines, $\delta$s, correspond to host mortality rates. Unlike *LT50*s, we find that the slopes are the same for the different initial pathogen densities, in the sense that we did not observe statistically significant differences between the mean mortality rates (one-way ANOVA, $F(2,15) = 0.2$, $P = 0.8$). The observed mortality rate is $\delta$ 0.055/h, corresponding to 70% of the host population dying every day. Our experimental observation that the mortality rate in the

exponential phase is independent of pathogen densities suggests that the mortality rate reflects the intrinsic lethality of the pathogen.

We next tested whether pathogen-induced mortality with (eventual) constant rate occurs ubiquitously across pathogens. We repeated the experiment under identical conditions but using pathogens *P. aeruginosa* (*Pa*), *S. marcescens* (*Sm*), and *S. enterica* (*Se*; Fig 1B). All survival curves exhibited an exponential phase, although the slopes of the lines are different for the three pathogens. In *Pa*, we confirm the result found in the previous experiment, whereas in *Sm* and *Se*, we find $\delta_{Sm}$ 0.02/h (~40% population death rate per day) and $\delta_{Se}$ 0.03/h (50%). We also confirmed that the lethalities $\delta_{Sm}$ and $\delta_{Se}$ are independent of the lawn pathogen density (Fig EV1), as we already showed for $\delta_{Pa}$. These results indicate that exponential death occurs in our experimental system for many different pathogens and that the lethality $\delta$ is a characteristic indicator of the host–pathogen interaction.

We note that ranking the pathogens by their lethalities is not consistent with the ranking obtained by their *LT50*s (Fig 1B inset). In fact, $50_{Sm}$ 70 h and $50_{Sm}$ 120 h suggest that *Sm* is more virulent than *Se*, whereas the slopes $\delta_{Sm}$ and $\delta_{Se}$ indicate the converse. This discrepancy arises because to fully understand survival curves, we also need to consider the time taken to enter the exponential phase (henceforth denoted by $\tau$), in addition to the lethality $\delta$. Indeed, *LT50* is strongly correlated to the time $\tau$, whereas the lethality $\delta$ is not (Fig 1C). In *Se*, the time required to enter exponential phase is twice the time in *Sm* ($\tau_{Se}$ 102/h and $\tau_{Sm}$ 50/h), although the exponential phase in *Se* is characterized by a sharper decline ($\delta_{Se} > \delta_{Sm}$). This signifies that *Sm* kills the hosts with a higher rate than *Se* at the early stages of the infection but is then surpassed by *Se* as the infection progresses. Therefore, the indicator pair ($\tau$, $\delta$) provides a more comprehensive description of the host survival curves than *LT50*.

### Theoretical model disentangles pathogen invasiveness and lethality from the survival curve of host population

To explain the previous results, we use a simple population dynamics model that incorporates a pathogen colonization rate *c*, pathogen growth rate *r*, and saturating population size *K* within the host. We assume that the pathogen population size within a worm, denoted by *N*, follows

$$\frac{dN}{dt} = (rN + c)\left(1 - \frac{N}{K}\right) \tag{1}$$

This simple model ensures that the pathogen population saturates at carrying capacity *K* (as compared to the similar choice for the right-hand side of equation (1) $rN(1–N/K) + c$, but this choice does not lead to any significant difference in our conclusions). We also assume that host mortality is linearly proportional to the pathogen load. In this case, the fraction of worms surviving $w(t)$ will change according to:

$$\frac{dw}{dt} = -\delta w \frac{N}{K} \tag{2}$$

where the constant $\delta$ is the lethality of the pathogen at saturation. Although models of wild disease are often sophisticated (Gog *et al*, 2015), we find that this exceedingly simple model suffices in our

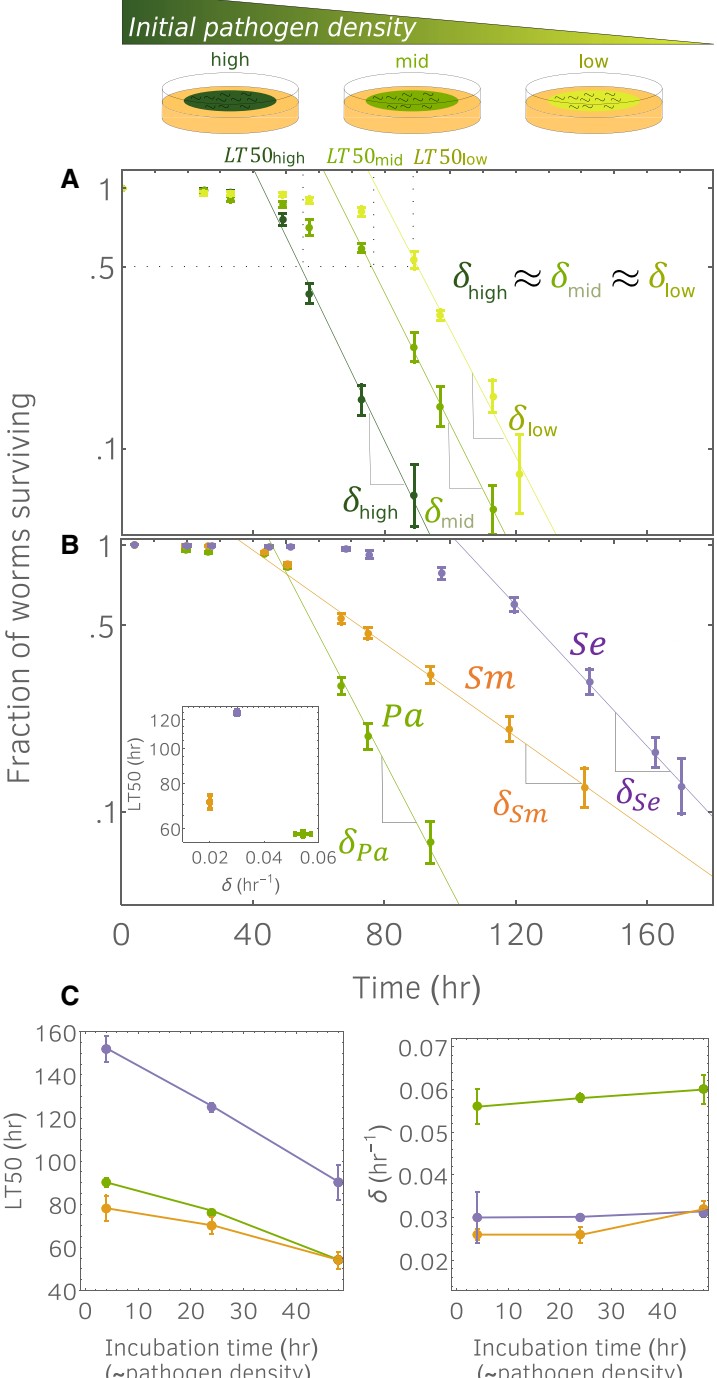

**Figure 1.  Host survival curves enter an exponential death phase (visualized as a line in semi-log scale), whose slope δ is characteristic of pathogen lethality.**

A   Survival curves obtained by exposing a population of *Caenorhabditis elegans* nematodes to pathogen *Pseudomonas aeruginosa*. Shade of green corresponds to different initial pathogen densities (high, mid, and low), obtained by pre-incubating the pathogen lawn for different times prior to adding the hosts. Markers correspond to experimental data averaged across six technical replicates. Error bars correspond to standard errors. Survival curves enter an exponential phase with a slope independent of the initial pathogen densities.

B   Survival curves obtained for bacteria *P. aeruginosa* (*Pa*), *S. marcescens* (*Sm*), and *S. enterica* (*Se*). For each pathogen, we report an exponential phase with a characteristic lethality (inset): *LT50* does not correlate with the lethality δ. Markers correspond to experimental data averaged across six technical replicates. Error bars correspond to standard errors.

C   *LT50s* (left) and δs (right) of different pathogens (colors matched panel B and Fig EV1), are plotted against the incubation time of the pathogen lawn. Unlike *LT50*, the lethality δ does not change with the initial pathogen density. Markers correspond to experimental data averaged across six technical replicates. Error bars correspond to standard errors.

Source data are available online for this figure.

setting. Moreover, equations (1) and (2) with our initial conditions can be exactly solved and yield predictions for the survival curve $w$ and the pathogen load $N$ in terms of the pathogen colonization rate, growth rate, and lethality (see the Appendix).

Our model predicts that the survival curves enter an exponential phase after a time $\tau$, as the pathogen load reaches carrying capacity $K$ (Fig 2). When $N(t) = K$, equation (1) reduces to $\dot{w} = -\delta w$, indicating that the per capita death rate, $\dot{w}/w$, is given by the constant $\delta$. The time taken to enter the exponential phase $\tau$, which we refer to as the invasion time, can be formally defined as the abscissa of the intersection point between the lines $w = 1$ and the asymptote of the exponential phase (see Fig 2). The invasion time can be written as:

$$\tau = \frac{1}{r}log(1 + K\frac{r}{c}). \tag{3}$$

We note that the invasion time $\tau$ does not depend on the lethality $\delta$, but rather provides a mechanistic summary of the pathogen's ability to invade the host, which is a combination of the ability to colonize and grow within the host. Indeed, a prediction of our model is that the time that it takes for the host mortality curves to reach the exponential phase (invasion time) corresponds also to the time that it takes the pathogen population to reach saturation within the host, which we will refer to as the saturation time. The quantity $\tau^{-1}$ therefore measures the pathogen invasiveness.

### Survival curves enter exponential phase as the pathogen load is at carrying capacity

Next, we tested the model prediction that the time taken to enter the exponential phase corresponds to the pathogen load growing to carrying capacity in the host population. The model also provides expressions for the survival curve and the pathogen load curves in terms of the pathogen colonization rate $c$, the growth rate $r$, and the lethality $\delta$ (see the Appendix). Fitting the theoretical predictions to experimental data would indicate that these three parameters determine the host–pathogen dynamics.

Our initial objective was to measure the pathogen load at a certain point in time. To do so, we collected approximately 10 worms, washed their cuticles to remove the external bacteria, grounded the sample population using a motorized pestle, and finally estimated the content of their intestines by colony counting (Fig 3A). Following this protocol, we measured the in-host growth curves for the three pathogens, the survival curves of which we already showed in Fig 1B (Fig 3B). After exposing the hosts for a day to a fully grown lawn, we measure significantly different number of cells: $N_{Se}(24\ h)\ 5 \times 10^3$ cells, $N_{Sm}(24\ h)\ 10^4$ cells, and $N_{Pa}(24\ h)\ 2 \times 10^4$ cells. This variation at early times reflects the varied colonization abilities of the pathogens, such as their survival rates as they pass through the grinder, the *C. elegans* tooth-like structure that crushes most bacteria prior to digestion (Cook, 2006). After a few days, all pathogen loads saturated to carrying capacity which differs up to an order of magnitude: $K_{Se}\ 1.7 \times 10^6$ cells, $K_{Sm}\ 1.1 \times 10^5$ cells, and $K_{Pa}\ 2.8 \times 10^5$ cells. We note that *Se* is the slowest colonizer but reaches the largest carrying capacity. In general, we found that the three pathogens exhibit different colonization abilities and carrying capacities within the hosts.

We then normalized the pathogen load curves by their carrying capacities to visualize these curves against the survival curves. In this way, we could test whether the entrance in the exponential phase occurs as the pathogen load curve plateaus (Fig 3C). We confirm that this is the case for the three pathogens: The times taken to enter exponential phase, $\tau$, were markedly distinct in the three cases

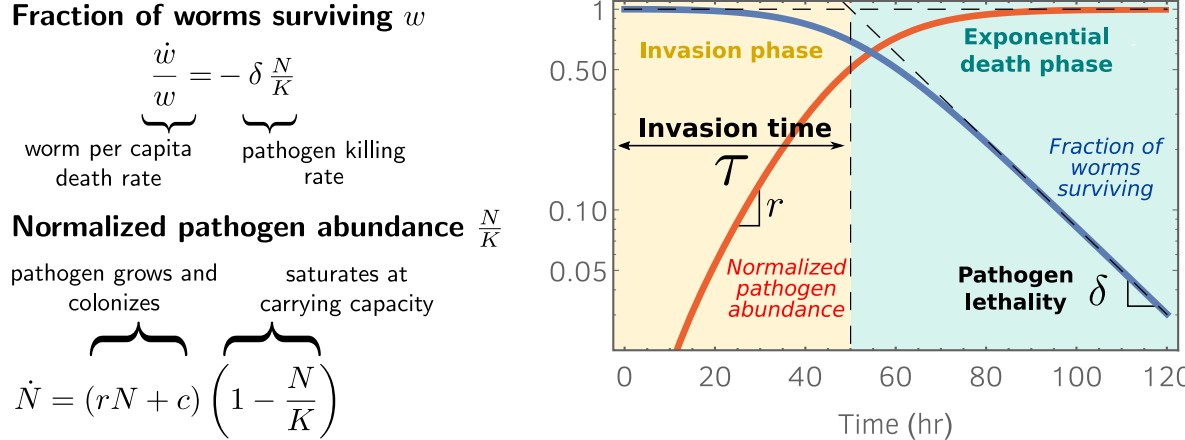

**Figure 2. A theoretical model predicts that the survival curves enter the exponential phase as the pathogen abundance inside the hosts (i.e., pathogen load) reaches carrying capacity.**

Our model formulates predictions for the survival curve $w$ and the pathogen load $N$, starting from the pathogen growth rate inside the host $r$, the pathogen colonization rate $c$, the pathogen lethality $\delta$, and the pathogen-carrying capacity $K$. We use the initial conditions $N(0) = 0$ and $w(0) = 1$. The right panel shows the fraction of worms surviving (solid blue line) and the per capita pathogen load normalized by the carrying capacity, $N/K$ (solid red line). The model disentangles the invasion phase, in which the worm mortality rate increases over time, from the exponential phase, where host mortality occurs with constant (maximum) rate $\delta$. The time taken to enter the exponential phase $\tau$ is given by the intersection between the exponential phase asymptote (diagonal dashed line) and $w = 1$ (horizontal dashed line).

Source data are available online for this figure.

(from 50 to 110 h), yet the invasion times are approximately equal to the saturation times, as predicted by our model (Fig 3D). Indeed, we observed that the pathogen load curves neatly plateaued as the survival curves reached the regime of constant death rate (Fig 3C). We tested whether our model quantitatively fit the pathogen load and survival curve. We already measured the death rate $\delta$ and carrying capacity $K$ for the three pathogens, and we determined the colonization rates $c$ and the growth rates $r$ by inspection. We found similar values for the growth rates of the three species ($r \sim 0.08$/h), whereas the colonization abilities of $Pa$ and $Sm$ (~250 cells/h) are better than $Se$ (~40 cells/h). The agreement between theory and data indicates that pathogen-induced mortality increases approximately linearly with pathogen load. We note that this did not have to be the case, as the pathogens could express virulence factors at any time during the infection process. Moreover, a simpler model in which host mortality is constant after exposure to the pathogen is not consistent with our data, as it could not explain the delay before the onset of host mortality (Fig 1A).

### Invasion time $\tau$ can be separated into colonization and replication time intervals

Next, we use our model and experiments to show that the invasion time $\tau$ can be split between a time period that is colonization-dominated and a time period that is replication-dominated: $\tau = \tau_c + \tau_r$ (Fig 4A). In the first-time interval, $\tau_c$, the pathogen influx is mostly due to external colonization rather than replication, since the hosts are initially sterile; in the second interval, $\tau_r$, colonization becomes negligible and the internal pathogen growth is the dominant effect. From our model, equation (2), we can show that (see the Appendix).

$$\tau_c = \frac{1}{r} log(2), \tag{4}$$

which, interestingly, shows that the time interval in which colonization dominates corresponds to the pathogen doubling time and is therefore independent of the colonization rate (although we have assumed $r \gg 4$ in the derivation of equations (3) and (4), which is always the biological case). This result means that the time in which pathogen abundance is dominated by colonization is equal to the pathogen replication rate. During time $\tau_c$, the pathogen abundance grows until it reaches $N^* = c/r$, after which replication becomes dominant. The invasion time $\tau$ can be rewritten as $\tau = \tau_c + \tau_r = r^{-1}$ log $(1 + K/N^*)$ which also shows that $\tau_r$ depends on $N^*$. Therefore, the time in which colonization dominates is largely independent of the lawn pathogen density (which, in our experiment, determines the colonization rate $c$).

From our data, we can estimate the colonization time scale $\tau_c$, the replication time scale $\tau_r$, and the killing time scale $\delta^{-1}$, for the three pathogens (Fig 4B). Since the pathogens have similar growth rates, the colonization time intervals $\tau_c$ are approximately equal. In contrast, $\tau_c$ (and hence $\tau$) is much larger in $Se$ compared with those of $Pa$ and $Sm$, due to the difference in carrying capacity. The killing time scale of $Se$ ($1/\delta$) is greater than the killing time scale of $Sm$ which, again, is due to the difference in carrying capacity (in fact, $Sm$ has a greater lethality per cell). It is also worth noting that the

$LT50$s for these three pathogens fall in the killing time interval (which follows the invasion phase), meaning that more than 50% of the hosts die in the exponential phase.

Finally, we inquired whether our model retains its predictive power when the colonization rates are varied experimentally. We analyzed the $Pa$ pathogen load curves obtained for different pathogen incubation times (Fig 1A) and found that, as expected, only the colonization rate $c$ needs to be varied to fit the three curves (Fig 4C). These results illustrate how the dynamics of pathogen colonization and growth are determined by the underlying processes by which the pathogen invades the host. In addition, the colonization rate c also depends upon the specific host–pathogen pair. For example, it depends on the survival probability of a pathogen cell as it goes through the grinder, the *C. elegans* structure that grinds bacterial cells prior to digestion.

## Discussion

In this study, we have demonstrated that integrating quantitative analysis of survival curves with mathematical modeling allows one to determine how the dynamics governing pathogen invasion of the host lead to the different time scales associated with host mortality. In particular, we find that the pathogen invasion time $\tau$ and lethality $\delta$ provide a better assessment of virulence than using $LT50$. The invasion time $\tau$ can be thought of as the time taken by the pathogen to reach carrying capacity in the host population. After such time, the host population dies with a constant rate that is given by the lethality $\delta$. For both indicators, we have formulated mathematically precise definitions (see Fig 2 and sub-section B of the Results section). Extracting $\tau$ and $\delta$ from the survival curves allows us to disentangle whether the salient pathogen characteristic is to be a good invader or a good killer, which is not possible to determine just by using $LT50$. These indicators also inform us about the shape of the survival curve according to the pathogen attributes. Skillful invaders exhibit a survival curve that drops rapidly into the exponential phase. Lethal pathogens are characterized by a survival curve that sharply declines once the pathogen reaches carrying capacity, suggesting that the host mortality rate is greater in the late stages of the infection. These results might lead to novel insight in studies in which various pathogens are screened (e.g., in *C. elegans*; Diard *et al*, 2007), and contribute to demonstrate that a reductionist approach to infection is possible (Hall *et al*, 2017). They also show that *C. elegans* is an excellent model system for unraveling simple quantitative laws in biology, as already recently proved in other fields such as aging (Stroustrup *et al*, 2016) and eco-evolutionary dynamics (Thutupalli *et al*, 2017).

Disentangling virulence into its causal attributes is necessary to understand the ecology and evolution of host–pathogen systems. Plant–pathogen systems provide the longest standing example that pathogens excel at a certain attribute in spite of others (e.g., Meyer *et al*, 2010), an effect that results from co-evolution (Fineblum & Rausher, 1995). Indeed, hosts can cope with pathogens by diminishing their invasiveness (host resistance) or by decreasing their lethality (host tolerance; Medzhitov *et al*, 2012). Such trade-offs have been observed in plants (Fineblum & Rausher, 1995) and animals (Råberg *et al*, 2007). However, there was no obvious trade-off between lethality and invasiveness among our three pathogens

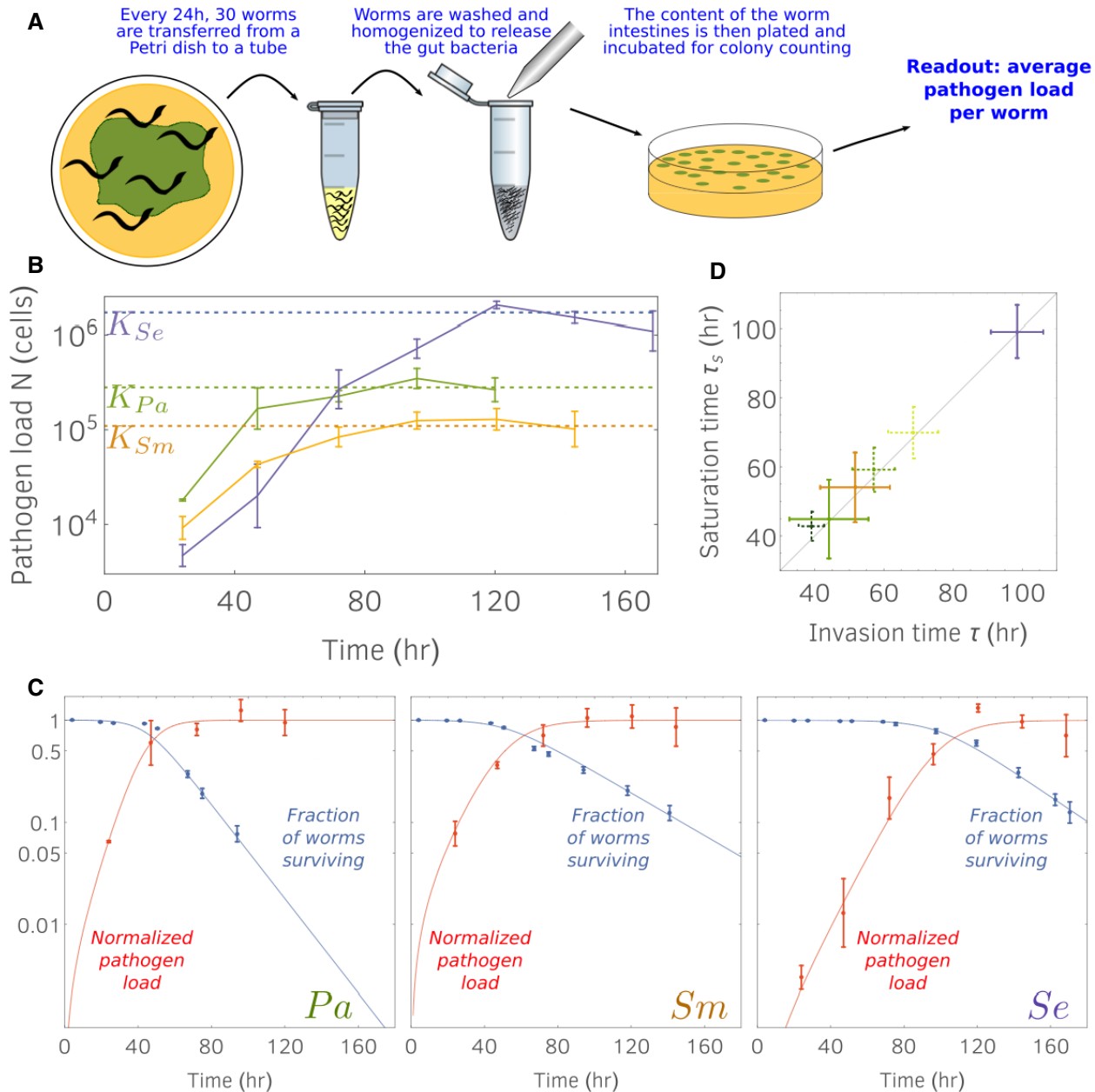

**Figure 3. Experiments confirm that hosts enter the exponential death phase at the same time that the pathogens saturate within the host.**

A Illustration of the protocol used to measure the pathogen load within the hosts.

B Per worm pathogen growth curves. Pathogen abundances data are obtained by averaging 3 or 4 replicates, each replicate consisting of a population of 10 worms. Error bars denote standard errors; dashed lines correspond to estimated pathogen-carrying capacities.

C Pathogen abundance data normalized by their carrying capacity (red markers) are plotted against the survival curve data (blue markers) of Fig 1B. Data are fitted by our model predictions (solid red and blue curves; parameter values in Dataset EV2; see Materials and Methods section for fitting procedure). 6 technical replicates, error bars are standard errors.

D As predicted by our model, the time taken by the pathogen population to saturate within the hosts is equal to the time necessary for the hosts to enter the exponential death phase (invasion time defined in Fig 2). Solid lines: saturation and invasion times estimated from data in Fig 1B and panel (B). Dashed lines: times estimated from data in Figs 1A and 4C. 6 technical replicates, error bars are standard errors.

Source data are available online for this figure.

(Figs 4B and EV2-left). Host resistance ($K^{-1}$) and tolerance ($K\delta^{-1}$), defined as in Råberg *et al* (2007), show that hosts are more resistant to *Sm* and *Pa*, but they have higher tolerance for *Se* (Fig EV2-right). These two indicators, however, do not account for how fast the pathogen invades, as they depend on $K$ and $\delta$, but not on the pathogen colonization rate $c$ or the pathogen growth rate $r$. In contrast, our indicator $\tau$ provides a time scale for invasion, and might therefore be important for host survival and for the

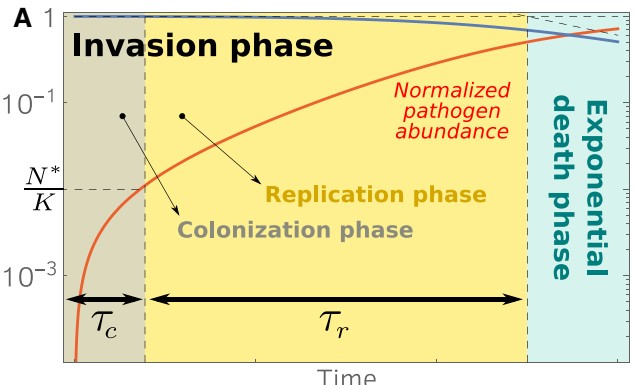

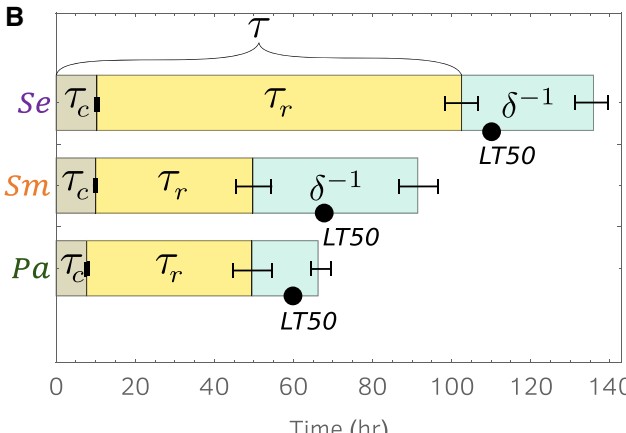

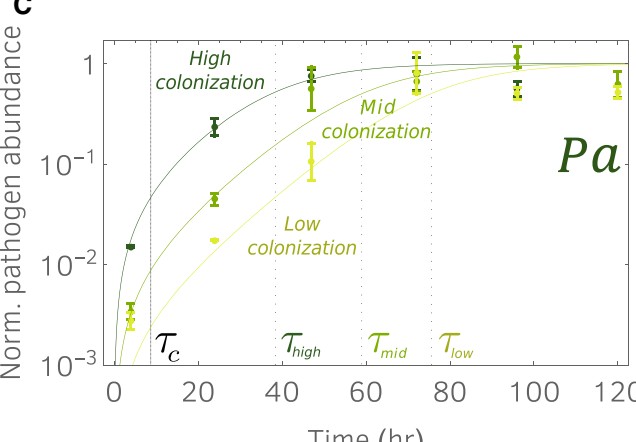

**Figure 4. Invasion phase can be further disentangled into a colonization phase and a replication phase.**

A   Invasion phase (see Fig 2) is initially dominated by colonization for a time $\tau_c = r^{-1}\ln 2$. As the pathogen load reaches $N = c/r$, replication is the leading effect for the remaining time $\tau_r$. The difference between colonization phase and replication phase results in a slope change in the normalized pathogen abundance (solid red line).

B   Time scales for the three pathogens (parameters values in Dataset EV2).

C   $Pa$ growth curves corresponding to survival curves shown in Fig 1A. Solid lines are model predictions for normalized pathogen abundances, obtained for same parameter values but different colonization rates (Dataset EV1). Invasion times for different colonization rates are shown. 6 technical replicates, error bars are standard errors.

Source data are available online for this figure.

epidemiology of a disease, especially in light of recent findings where pathogen strategies in human disease are disentangled (Regoes *et al*, 2014).

A key observation in this study is that survival curves enter an exponential death phase, namely the hosts eventually experience a constant mortality rate. The interpretation of this result is that the damage of virulence is "memoryless", in the sense that the damage does not accumulate with time. We expect that this finding will not hold in more complex animals *than C. elegans*, where more effects are in play (e.g., adaptive immunity). Although to our knowledge this is the first report in an experimental host–pathogen system, exponential laws are already well-established in many other areas of biology. It is textbook knowledge that microbes die exponentially after stress (for pH stress see *e.g.,* (Withell, 1942), for temperature stress (Peleg & Cole, 1998)) or prolonged starvation (Madigan *et al*, 2017), even though density-dependent deviations have been observed (Phaiboun *et al*, 2015), which can be due to cell memory effects (Mathis & Ackermann, 2016). The most frequent example of exponential statistics is perhaps found in the logistic growth of microorganisms, where the exponential face is usually preceded by a lag phase (Madigan *et al*, 2017). Interestingly, the expression that relates the lag time is remarkably similar to our equation (3) (Manhart *et al*, 2018), which highlights a parallel between microorganismal growth and host decay (Wang & Goldenfeld, 2010). In this respect, our pathogen lethality $\delta$ plays the analog of the bacterial growth rate, which has recently shown to be one of the few key fundamental parameters determining the state of the cell (Scott *et al*, 2010). Finally, obligatory parasites have been recently shown to decay exponentially in a biphasic fashion, when found outside the host (Brouwer *et al*, 2017).

We found that the decay in *C. elegans* populations reaches an exponential phase after exposure to three well-studied pathogens, although when the hosts are exposed to non-pathogenic bacteria such as *Pseudomonas chlororaphis* we find that there is no exponential phase (Fig EV3). Indeed, even after 10 days of feeding on *P. chlororaphis* the host mortality rate had not yet stabilized. In contrast, the bacterial load saturates after 3 days, remains stable for the successive 3 days, and then starts growing again. This effect might be due to the fact that the bacterial carrying capacity increases as the worm ages (Portal-Celhay *et al*, 2012). Further work will be required to clarify whether our framework has a more general applicability. With recent technological advances (Lee *et al*, 2016), it might be possible to adopt automatized protocols that would allow to repeat our investigations with higher throughput. Indeed, we corroborate our findings by analyzing survival curves of a *C. elegans* population from Levine Lab (Lee *et al*, 2016), where mortality is induced by a pathogen, and from Fontana's Lab (Stroustrup *et al*, 2016) in which instead senescence occurs (Fig EV4). The data from the Levine Lab, obtained using the same *C. elegans/P. aeruginosa* system but using a microfluidic chamber setting, seem to exhibit an exponential phase with an exponent consistent with our findings. In the case of age-induced death, the survival curve from Fontana's Lab exhibits (over time scale significantly longer than the ones monitored in our experiments) a non-linear dynamics that cannot be fully described by a single exponent. This is in agreement with the fact that senescence is thought to follow Gompertz's law (Abrams & Ludwig, 1995), even though a recent study suggests that, in humans, senescence might possess an exponential plateau (Barbi *et al*, 2018).

Finally, our investigations have been limited to single pathogens in sterile host populations, thus neglecting the interaction between the pathogen and the host microbiome. Recent works have demonstrated that *C. elegans* is a suitable model system for microbiome investigations (Vega & Gore, 2017; Zhang *et al*, 2017), and demonstrated the profound effects that gut commensal bacteria might have in fending off pathogen of affecting host life span (Clark & Hodgkin, 2014).

For example, it is known that nematodes previously colonized with certain microbes exhibit enhanced survivability during a pathogen infections (Portal-Celhay & Blaser, 2012; Montalvo-Katz *et al*, 2013; King *et al*, 2016). With our framework, we can quantify

how each pathogen attribute is affected as a function of the microbiome composition, thus formulating novel hypotheses for elucidating interspecies mechanics. In future work, it would be interesting to pre-colonize the host population with a commensal bacteria, before exposing the population to a pathogen. In this way, it would be possible to investigate how the invasion time $\tau$ and the lethality $\delta$ change in the pre-colonized population, thus revealing in a quantitative fashion the role of the microbiome in fending off the pathogen. It is our hope that simple quantitative laws of host–pathogen dynamics revealed by our system may provide insight into pathogenesis in more complex host–pathogen systems.

# Materials and Methods

## Reagents and Tools table

| Reagent/Resource | Reference or source | Identifier or catalog number |
| --- | --- | --- |
| **Experimental models** | | |
| PA14 (*P. aeruginosa*) | Ausubel's Lab (Harvard) | |
| Db10 (*S. marcescens*) | CGC | |
| LT2 (*S. enterica*) | CGC | |
| OP50 (*E. coli*) | CGC | |
| SS104 (*C. elegans*) | CGC | *glp-4(bn2)* |
| **Chemicals, enzymes, and other reagents** (e.g., drugs, peptides, recombinant proteins, and dyes) | | |
| Triton X-100 | Sigma-Aldrich | X100-5ML |
| Levamisole hydrochloride | Sigma-Aldrich | #1359302 |
| Gentamicin sulfate | Sigma-Aldrich | #1289003 |
| Carbenicillin disodium salt | Sigma-Aldrich | C1389 |
| Poly(ethylene glycol) methyl ether | Sigma-Aldrich | #81316 |
| **Software** | | |
| Wolfram Mathematica 11 | https://www.wolfram.com/mathematica/ | |
| R 3.5.2 | https://www.r-project.org/ | |
| **Other** | | |
| Kimble Kontes tubes | Grainger | # 6HAY2 |
| Pellet pestles | Sigma-Aldrich | Z359947 |
| Pellet pestles cordless motor | Sigma-Aldrich | Z359971 |

## Methods and Protocols

### Worm and bacterial strains

The bacterial strains used in the paper are *Pseudomonas aeruginosa* PA14 (from Ausubel's Lab, Harvard), *Serratia marcescens* Db10 (Caenorhabditis Genetics Center, CGC), *Salmonella enterica* LT2 (CGC), *Escherichia coli* OP50 (CGC), and *Pseudomonas chlororaphis* (ATCC 9446). Throughout the work, we used *Caenorhabditis elegans* strain SS104 (*glp-4(bn2)*) obtained from CGC. Due to the *glp-4* mutation, this strain is able to reproduce at 15°C but is reproductive sterile at 25°C; use of this strain prevented the worms from producing progeny during experiments, ensuring that the only changes in worm population were due to pathogen-induced mortality.

### Preparation of worm cultures

Synchronized (i.e., same age) worm cultures were obtained using standard protocols (Stiernagle, 2006). For propagation of worms, SS104 cultures were maintained at 15°C on NGM agar plates with lawns of the standard food organism *E. coli* OP50. For synchronization, worms from several nearly starved plates were washed with sterile distilled water and treated with a bleach-sodium hydroxide solution; the isolated eggs were placed in M9WB overnight to hatch, and then transferred to NGM + OP50 plates at the sterility-inducing temperature (25°C) for 2 days to obtain synchronized adults. Worms were then washed from plates using M9 worm buffer + 0.1% Triton X-100 (Tx) and then rinsed with M9 worm buffer. Worms were then transferred to S medium + 100 µg/ml gentamicin + 5X heat-killed OP50 for 24 h to kill any OP50

inhabiting the intestine, resulting in germ-free synchronized worms. To test for antibiotic efficacy, we grind the worm population (see Pathogen load assay sub-section) and plate the solution on three LB agar plates which we then incubate at 30°C for 48 h. Antibiotic is considered effective as we did not find any colony-forming units in agar plates. These 3-day-old synchronized adult worms were then rinsed in M9PG (M9WB + 0.1% PEG; PEG is used to prevent the worms from sticking to the pipette tip), washed via sucrose flotation to remove debris, and rinsed 3X in M9PG worm buffer to remove sucrose before use in experiments.

### Survival curves assay

To generate the data of Fig 1A and B, we used a variation of a previously published protocol (Kirienko *et al*, 2014). For a single condition (i.e., a survival curve on *Pa*), we run our assay in a 6-well plate where each well (4 cm diameter) represents a technical replicate. Each well is filled with 4 ml of SK agar media (recipe in [16]). To prevent worms from exiting the plate, we add 15 μl of palmitic acid (10 mg/ml in EtOH) to each well border. Pathogen monocultures are grown for 24 h at 30°C in 5 ml LB, after which the monocultures reach saturation. Then, 7 μl of culture is pipetted to the center of each plate and spread using a small metal cell spreader to create a pathogen lawn. The 6-well plate is then parafilmed to prevent evaporation and incubated at 25°C for the desired time (Fig 1A: 4 h, low density; 24 h, mid-density; 48 h, high density. Figure 1B: 48 h). To each well, we then add off-lawn a population of ~50 adult reproductive sterile germ-free worms suspended in M9PG, as described in the previous section. As the buffer is quickly absorbed by the plate, the worms start feeding on the pathogen lawn. We keep the 6-well plate parafilmed and incubated at 25°C throughout the whole experiment, and we monitor the number of worm surviving using standard worm picking protocols (Kirienko *et al*, 2014).

### Pathogen load assay

Our protocol is a variation of previously published protocols (Alegado *et al*, 2003; Portal-Celhay & Blaser, 2012; Portal-Celhay *et al*, 2012). Briefly, we use two buffers: TXLV (M9WB with 1% Triton-X + 50 mM levamisole) and TXAB (TXLV + gentamicin and carbenicillin ~210 μg/ml). We prepare 6-well plates as described in the previous section. Since measurement of the pathogen load is destructive, we could not use the same worm population to measure both the pathogen load and the survival curve. Thus, we prepared a group of plates under identical conditions and separated them into two groups: We estimated the survival curve from one group and the pathogen load from the other, thus assuming that the two groups possess equal average dynamics. To measure the pathogen load, we collected each day 30 alive worms by washing the plates with M9WB. We then washed worm resuspension, four times in TXLV. Due to levamisole, worm peristalsis is interrupted in a few seconds, and the mouth and anus of the worm remain shut, thus preventing the internal bacteria to be flushed out. We then resuspended the worm population in TXAB and incubated for 1 h at 25°C with gentle shaking. Every 20 min, we washed and resuspended the population in fresh TXAB. The purpose of the incubation is to remove the external bacteria attached to the worm cuticle. We then washed the worms 3–4 times in TXLV to remove the antibiotic and transfer the population to a small Petri dish. Using a dissecting scope, we removed the worms that were not fully paralyzed. We then split

the worm population into three Kontes tubes so that each tube contains 50 μl TXLV and 10 worms. Each tube constitutes a technical replicate. We homogenize the worms in a tube with a pellet pestle for 1 min continuously. We then dilute the solution in M9WB, and plate to LB agar plates for colony counting. Pathogen load is reported as the mean ± SEM of the three technical replicates.

### Statistical analysis

Details of our statistical analysis are provided as a supplementary *R* notebook.

Markers in Fig 1A and B represent mean survival curve averaged across 6 technical replicates. Corresponding *LT50*s were determined graphically from mean survival curve. Solid lines in Fig 1A and B were obtained by fitting the mean survival curve using a linear model. The consistency of the mortality rates in Fig 1A (and analogous figures, Fig EV1) has been tested using one-way ANOVA (results in main text).

In Fig 1C, we reported lethalities of different pathogens, estimated for different initial densities. The corresponding survival curves are shown in Figs 1A and EV1. To estimate lethalities, we determined the fitting region from the mean survival curve, and then fitted each technical replicate using a linear model. Next, we computed average lethality and their standard errors obtaining the values in the tables below. These values are consistent with those in Dataset EV1, which were obtained with a non-linear fit, as described later.

|  | *Pa* (high) | *Pa* (mid) | *Pa* (low) |
|---|---|---|---|
| Average lethality δ(/h) | 0.057 | 0.053 | 0.052 |
| Standard error (/h) | 0.009 | 0.003 | 0.006 |

|  | *Sm* (high) | *Sm* (mid) | *Sm* (low) |
|---|---|---|---|
| Average lethality δ(/h) | 0.034 | 0.031 | 0.028 |
| Standard error (/h) | 0.005 | 0.004 | 0.004 |

|  | *Se* (high) | *Se* (mid) | *Se* (low) |
|---|---|---|---|
| Average lethality δ(/h) | 0.035 | 0.031 | 0.034 |
| Standard error (/h) | 0.005 | 0.003 | 0.005 |

In Fig 1B, we repeated the experiment with a single incubation time but using 12 technical replicates for each pathogen. We estimated lethalities following same procedure described above. We find the following values:

|  | *Pa* | *Sm* | *Se* |
|---|---|---|---|
| Average lethality δ(/h) | 0.058 | 0.022 | 0.033 |
| Standard error (/h) | 0.004 | 0.002 | 0.006 |

Markers correspond to mean pathogen loads averaged over 3–4 technical replicates obtained for different initial pathogen densities. Pathogen loads are rescaled by the carrying capacity $K_{pa} = 2.8 \times 10^5$ cells, which we determined by inspection. The mean pathogen loads

are fitted using the solution of equation (1) (see the Appendix). To carry out the fitting procedure, we imposed the same growth rate $r$ for the three conditions, but different colonization rates: $c_{high}$, $c_{mid}$, and $c_{low}$. For each condition, we computed the sum-of-square error between the solution of equation (1) and the mean pathogen load data. We estimated the four parameters $r$, $c_{Pa,48}$, $c_{Pa,24}$, and $c_{Pa,4}$ by minimizing the sum of the errors for the three fits. To provide errors for the fitted parameters, we bootstrapped the pathogen loads data and repeated the fitting procedure, hence obtaining different values for $r$, $c_{Pa,48}$, $c_{Pa,24}$, and $c_{Pa,4}$. We repeated the fit 500 times and computed the standard deviations of the four parameters, which represent our errors on the fitted data. Values and errors are shown in the Dataset EV1.

Figure 1B show mean pathogen load data averaged over 3–4 technical replicates for each pathogen. These curves are used to determine (by inspection) the carrying capacities (see Dataset EV2). We then fit the model solutions of equations (1) and (2) (explicit formulae in the Appendix) to the normalized pathogen load data and the corresponding survival curves in Fig 1B. Our fitting procedure is similar to that in Fig 4C. We estimated the parameters $r$, $c$, and $\Lambda$ $\delta$ for the three pathogens by minimizing the sum of the square errors for the pathogen load and the growth curve. Errors to these parameters are given by bootstrapping. Values and errors are shown in the Dataset EV2.

## Data availability

Raw data and R notebook containing the statistical analysis are available at https://github.com/lewlin/disentangling-pathogens-SM.

**Expanded View** for this article is available online.

### Author contributions
TB and JG designed the study. TB carried out the experiments, analyzed the data, and performed a mathematical analysis of the model. TB and JG wrote the paper.

### Conflict of interest
The authors declare that they have no conflict of interest.

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
