## [Review Process File · Molecular Systems Biology]

Disentangling bacterial invasiveness from lethality in an experimental host-pathogen system

Tommaso Biancalani and Jeff Gore.

Review timeline:	Submission date:	29 th October 2018
	Editorial Decision:	8 th January 2019
	Revision received:	1 st April 2019
	Editorial Decision:	2 nd May 2019
	Revision received:	15 th May 2019
	Accepted:	17 th May 2019

Editor: Maria Polychronidou

Transaction Report:

1st Editorial Decision

8th January 2019

Thank you again for submitting your work to Molecular Systems Biology. We have now heard back from the three referees who agreed to evaluate your study. As you will see below, the reviewers acknowledge that the presented findings seem interesting. They raise however a series of concerns, which we would ask you to address in a major revision.

Without repeating all the points listed below, some of the most fundamental issues are raised by reviewer #3 who thinks that further controls and validations need to be included to better support the conclusions of the study.

All other issues raised by the reviewers need to be satisfactorily addressed. As you may already know, our editorial policy allows in principle a single round of major revision so it is essential to provide responses to the reviewers' comments that are as complete as possible. Please feel free to contact me in case you would like to discuss in further detail any of the issues raised by the reviewers.

REFeree REPORTS.

Reviewer #1:

Review of the manuscript by Biancalani and Gore

Summary of the work

Biancalani et al probe the temporal dynamics underlying the survival curve of a host infected by a virulent parasite. They use *C. elegans* and its bacterial parasites as a model system. They found that

the commonly used temporal measure of LT50 (time to 50% death) is strongly dependent on environmental conditions and does not represent well the full temporal dynamics of the survival curve and its underlying mechanisms.

Instead, they show using a combination of modeling and measurements of bacterial loads and worm death, that the survival curves are characterized by two time-scales corresponding to two phases - an invasion phase which corresponds to the duration it takes the bacteria to infect and grow to carrying capacity and a lethality phase, which is characterized by an exponential decay in survival with a characteristic virulence time-scale. At a higher resolution, they further show that the invasion time can be divided to a colonization time (equal to the doubling time of the population) and a growth phase, from the initial colonizers to the carrying capacity.

Importantly invasion time-scale and virulence time-scale can vary independently and the three bacterial species they observe have different characteristics of these two time-scales. The two time-scales also depend differently on environmental parameters, such as bacterial density in the environment, which strongly affect invasion time with no effect on the virulence death rate. The LT50 measure is mostly affected by invasion time and not much by virulence time-scale per-se, making it a problematic measure.

Assessment

General

I find the manuscript to be interesting and enlightening in the way it decomposes a single (crude) measure of temporal dynamics into several refined measures which shed light on different mechanistic aspects underlying the dynamics. I think the results are self-contained and demonstrate nicely the validity and utility of this decomposition. My main comments concern additional discussion points concerning the generality of some of the findings and possible mechanisms explaining them, as well as a suggestion for a simple experiment that will increase the visibility of the work, but is not necessary for its acceptance.

The work will be interesting for people the fields of epidemiology, microbe-host interactions and others.

Specific comments:

1. The first two comments concern the generality of the finding and specifically the somewhat surprising observation of exponential decay in survival rate.

a. the fact that death rate is constant at longer times, imply a one-step memory-less death mechanism. This is kind of surprising, as it implies that damage of virulence is not accumulated with time, but remains in a kind of a (unhealthy) steady-state. This should be directly discussed in the discussion section and mechanisms that lead to such behavior or contrary - to a non-exponential behavior - should be mentioned.

b. Besides bacterial death, there is a wealth of data on survival curves in humans for viral infection (e.g., HIV) or cancer. Are there no other cases of long-term exponential decay, or is this typically uncharacterized and researchers only measure LT50. Note that if one measures LT50 and LT99 (time to 99% death), then one can extract the two time-scales discussed here easily. It would be enlightening if the authors would be able to re-analyze previously published survival curves using their decomposition. I realize that this is a daunting task, as this data may not be readily accessible. I don't think that answering this comment is necessary for publication, as much as that it would increase the visibility of the work.

2. Another easy experiment that may increase the visibility of the work (but is not strictly necessary), is to show how the two time-scales are affected in cases where there is a clear change in LT50. One easy experiment is to compare the invasion time and virulence time in aged worms (1 week old) vs. young worms (as was done here). It is known that the LT50 of aged worms is reduced,

but what is the effect on virulence time-scale? Again - I do not require this for publication, but think that this would increase the visibility of the work.

Reviewer #2:

Disentangling bacterial invasiveness from lethality in an experimental host-pathogen system
Biancalani and Jeff Gore
Mol. Syst. Biol.

Summary

In this work, the authors argue that the commonly used metric for quantifying bacterial virulence, the time taken to kill 50% of infected hosts (LT50), is insufficiently informative of different modes of bacterial virulence. They propose separating virulence into metrics of bacterial lethality and invasiveness, both of which can be extracted from the survival curve. They further separate invasiveness into two regimes, governed by colonization and pathogen growth.

General comments

This is an excellent study. Through well-designed, thorough quantitative analysis, it highlights some limitations of some current metrics in quantifying bacterial pathogenesis. Moreover, the presented phenotypic quantifications provide a starting point for future studies aiming at developing a more mechanistic understanding.

One of the most interesting aspects of the work is the constant death rate (of the host), when a bacterial pathogen reaches the carrying capacity. This death rate is largely independent of the bacterial inoculum size. The authors did not attempt to provide a mechanistic explanation for this feature. I understand the advantage in coarse-grained analysis (by modeling and experiment) for the questions that the authors ask. However, I very much hope that the authors would discuss the potential mechanistic basis for their results. For instance, is this behavior a result of quorum sensing? It is well established that QS plays a central role in the pathogenesis of different bacteria, particularly in *P. aeruginosa*. Given their results (particularly the independence of inoculum size), the possibility of QS involvement is rather tempting. Some discussion on this point would be helpful.

From my first impression, their model formulation (on bacterial growth) is somewhat unusual. It appears that it might be more consistent with their experimental protocol by (1) using a simple logistic equation (without "c"), and (2) assuming a positive initial density, corresponding to the inoculum. This question is relevant as it seems that the dissection of the invasion time into colonization time (τ_c) and replication time (τ_r) is tied to the model formulation. By eyeballing, it appears that the experimental data can be equally well fit by using a simple logistic equation, where the initial phase of the growth curve is not bent (Figure 3).

Other specific comments

1. The paragraph beginning on paragraph 270 seems to muddy the lethality/invasiveness being a property of both host and pathogen, since it establishes that "hosts can cope with pathogens by diminishing their invasiveness (host resistance) or decreasing their lethality (host tolerance)" but then goes on to say that "these two indicators [host resistance and tolerance]... do not account for how fast the pathogen invades," which is odd considering the parallel drawn between reduced invasiveness and increased resistance from the earlier sentence and also makes it seem like invasion speed is a property of pathogen and not host.

2. Aside from the technical issue associated with the dissection between colonization and replication times, but it's unclear what the utility of splitting the invasion time into these two phases is when it comes to generally understanding the host-pathogen interaction (as opposed to simply noting that colonization and replication rates, not times, affect the invasion time differently). Further

elaboration on this point is useful.

3. Nomenclature in the last section is somewhat confusing given that the colonization time is dependent on growth rate, not colonization rate as per equation (4), and Figure 4C shows growth curves with identical "colonization time" but whose invasion times vary with "colonization rate." I wonder whether this is the best approach to these designations but confess I do not have a good alternative since both are meant to indicate a physiological process.

4. Figure S1's caption has a few typos: currently reads "Figures S1" and "the LT50s is". The text box is also too small vertically, cutting off some of the text.

5. Line 245 parentheses should maybe read "which follows the invasion phase" instead if "which following"

6. Lines 104-105 make it clear that the metrics reflect both the pathogen and the host; it might be good to add this earlier (line 69-70, where it currently just says pathogen-specific).

Reviewer #3:

SUMMARY

The manuscript by Biancalani et al. focuses on modeling and quantifying different steps in bacterial virulence using the nematode *Caenorhabditis elegans* as a system. The authors propose to decompose bacterial virulence in two main phases: invasion and lethality. The authors define "invasion" as the time frame in which the bacterial pathogen colonizes its host and "lethality" (δ) as the time frame where the pathogen kills its host. The authors utilize *C. elegans* nematodes exposed to three different bacterial pathogens to develop a model of this process. The authors suggest that bacterial load linked to onset of lethality at specific timepoints that integrate the virulence and host susceptibility. However, the utility of this model is limited in its current form due to minimal effort in its validation, merging of independent host and pathogen variables and limited experimental details to facilitate its reproducibility.

MAJOR POINTS

1) The central premise is gleaned from a dilution experiment of *Pseudomonas*, but the authors fail to acknowledge other confounding factors in the experimental setup.

a. From the host perspective, thin lawns of bacterial food source have been shown to induce caloric restriction and alter immune / stress response pathways, plus could also change pumping and defecation rates that all could influence the rate of "invasion". Were any of these measures examined?

b. From the pathogen perspective, virulence factors in all of these pathogens are regulated by quorum sensing and cell density. Thus, each lawn may have dramatically different levels of toxins. Was this measured in any way?

c. Similarly, It is unclear how the authors link bacterial lawn density to colonization rates and how it affects the model. It is unclear how the authors characterize the different fixed parameters such as the colonization factor c . Was this quantified?

2) Few variables are included and model lacks true validation set. It is unclear how the model keeps its predictive nature when settings are defined after the experiment. It would have been stronger and more convincing to test the predictivity of the model on a different set of pathogens as the ones used in the present study.

3) As promising as this works seems, the authors only use three bacterial pathogens that have very different virulence mechanisms. Additionally, only one non-pathogen is tested as a negative control, it would have been more informative to test more than the single non-pathogenic bacteria, pathogen virulence mutants (e.g., QS defective) or in more immune-compromised host strains (e.g., *pmk*-

1;glp-4). Further, in Figure S2, it is confusing why the authors claim that the model does not apply to non-pathogenic bacteria, as it seems that the curves appear to trend toward a similar pattern as the pathogens and the experiment was ended too soon to make that conclusion as few of the animals have started to die.

MINOR POINTS

4) Many of the details surrounding the central experiments could be better delineated. For example, the actual concentrations of bacteria used in any of the experiments is not noted, nor is it clear whether a comparable amount is used for each of the pathogens.

5) Microscopy-based (FISH or otherwise) visualization of bacterial loads would help ensure that the overall levels are reflective of the population and not due to a jackpotting effect by a few highly colonized animals.

6) Interpretation is limited by the fact that few / no bona fide (reagent) controls are included in the results. Were the antibiotics effective? How quickly were animals paralyzed at the different stages? Also, for one of the central assays (colonization), premixing of the 30 animals seems likely to normalize potential animal-animal variation that may bias interpretation. Lastly, it is not clear how the precise numbers of animals were transferred, whether the animal number was counted or rather inferred.

7) Similarly, non-normalized values for colonization levels should also be presented to ensure values are within the realm of published experiments.

8) A discussion of the relevance of these results to natural pathogens and commensals of *C. elegans* would be helpful for applicability.

1st Revision - authors' response

1st April 2019

RESPONSE TO REVIEWER 1

Summary of the work

Biancalani et al probe the temporal dynamics underlying the survival curve of a host infected by a virulent parasite. They use *C. elegans* and its bacterial parasites as a model system. They found that the commonly used temporal measure of LT50 (time to 50% death) is strongly dependent on environmental conditions and does not represent well the full temporal dynamics of the survival curve and its underlying mechanisms.

Instead, they show using a combination of modeling and measurements of bacterial loads and worm death, that the survival curves are characterized by two time-scales corresponding to two phases - an invasion phase which corresponds to the duration it takes the bacteria to infect and grow to carrying capacity and a lethality phase, which is characterized by an exponential decay in survival with a characteristic virulence time-scale. At a higher resolution, they further show that the invasion time can be divided to a colonization time (equal to the doubling time of the population) and a growth phase, from the initial colonizers to the carrying capacity.

Importantly invasion time-scale and virulence time-scale can vary independently and the three bacterial species they observe have different characteristics of these two time-scales. The two time-scales also depend differently on environmental parameters, such as bacterial density in the environment, which strongly affect invasion time with no effect on the virulence death rate. The LT50 measure is mostly affected by invasion time and not much by virulence time-scale per-se, making it a problematic measure.

Assessment

 General

I find the manuscript to be interesting and enlightening in the way it decomposes a single (crude) measure of temporal dynamics into several refined measures which shed light on different mechanistic aspects underlying the dynamics. I think the results are self-contained and demonstrate nicely the validity and utility of this decomposition. My main comments concern additional discussion points concerning the generality of some of the findings and possible mechanisms explaining them, as well as a suggestion for a simple experiment that will increase the visibility of the work, but is not necessary for its acceptance.

The work will be interesting for people the fields of epidemiology, microbe-host interactions and others.

We thank the referee for the nice summary and appreciate the enthusiastic response.

Specific comments:

1. The first two comments concern the generality of the finding and specifically the somewhat surprising observation of exponential decay in survival rate.

a. the fact that death rate is constant at longer times, imply a one-step memory-less death mechanism. This is kind of surprising, as it implies that damage of virulence is not accumulated with time, but remains in a kind of a (unhealthy) steady-state. This should be directly discussed in the discussion section and mechanisms that lead to such behavior or contrary - to a non-exponential behavior - should be mentioned.

We agree with the reviewer that the interpretation of our data is that the damage of virulence is not accumulated in time. We believe that this finding is intriguing, but we do not expect such behavior to hold in presence of additional layers of complexity (for example, adaptive immunity). We modified "Discussion" session in the main text by adding the following sentence:

LINE 298

The interpretation of this result is that the damage of virulence is "memoryless", in the sense that the damage does not accumulate with time. We expect that this finding will not hold in more complex animals than *C. elegans*, where more effects are in play (e.g. adaptive immunity).

b. Besides bacterial death, there is a wealth of data on survival curves in humans for viral infection (e.g., HIV) or cancer. Are there no other cases of long-term exponential decay, or is this typically uncharacterized and researchers only measure LT50. Note that if one measures LT50 and LT99 (time to 99% death), then one can extract the two time-scales discussed here easily. It would be enlightening if the authors would be able to re-analyze previously published survival curves using their decomposition. I realize that this is a daunting task, as this data may not be readily accessible. I don't think that answering this comment is necessary for publication, as much as that it would increase the visibility of the work.

We could not find high-quality data from humans from which we could extract time scales so as to make meaningful comparisons. In particular, in order to unambiguously determine whether human survival curves in the presence of a given pathogen display an exponential phase would require that the mortality approach unity (ie ~99% death) so that there is sufficient "range" to the data to address the point. We have however added a new figure in which we analyze survival curves of *C. elegans* populations obtained in two different labs (see Fig. EV4).

Figures EV4: Analysis of experimental data from two other studies suggests that exponential phase is observed in host-pathogen dynamics whereas a more complex scenario arises in senescence. (top) Survival kinetics for the *C. elegans*/*P. aeruginosa* system, measured using *HandKAchip* – Hands Free Killing Assay on a Chip (data from Erel Levine’s lab, 2016). The survival kinetics exhibits an exponential phase, consistently with our prediction, with exponent $\delta \sim 0.06 h^{-1}$. **(bottom)** Survival kinetics of *C. elegans* wild-type population due to senescence from Walter Fontana’s lab (2013). The survival kinetics, observed for a much larger time range than in our experiments, decay in a non-linear way, which is approximately exponential when restricted to certain time sub-intervals (e.g. two linear fits shown in figure). For age induced-death, the overall death rate of the host population is smaller than those observed in our pathogen-induced mortality experiments.

We confirm that in the dataset from the Levine’s lab, where the worm population is exposed to a pathogen, the survival curve displays an exponential phase consistent with our findings. In contrast, the dataset from the Fontana’s lab obtained with the *C. elegans* life span machine does not seem to be fully characterized by a single exponential phase (consistent with the fact that death is not induced by aging and not by a pathogen). We added the following paragraph in the “Discussion” section:

LINE 326

The data from the Levine lab, obtained using the same *C. elegans*/*P. aeruginosa* system but using a microfluidic chamber setting, seem to exhibit an exponential phase with an exponent consistent with our findings. In the case of age-induced death, the survival curve from Fontana’s lab exhibits (over time scale significantly longer than the ones monitored in our experiments) a non-linear dynamics that cannot be fully described by a single exponent. This is in agreement with the fact that senescence is thought to follow Gompertz’s law (Abrams & Ludwig, 1995), even though a recent study suggests that, in humans, senescence might possess an exponential plateau (Barbi *et al*, 2018).

2. Another easy experiment that may increase the visibility of the work (but is not strictly necessary), is to show how the two time-scales are affected in cases where there is a clear change in LT50. One easy experiment is to compare the invasion time and virulence time in aged worms (1 week old) vs. young worms (as was done here). It is known that the LT50 of aged worms is reduced, but what is the effect on virulence time-scale? Again - I do not require this for publication, but think that this would increase the visibility of the work.

We agree that it would be interesting to study how the time scales that we have identified change as we alter the host (age, immune activity, grinding, peristalsis), the pathogen (expression of virulence factors), and the environment (temperature, media, etc). However, given the extensive nature of this list—and the range of experimental conditions that we have already explored—we believe that it is best to publish our results now and to leave these other questions to future work.

Reviewer #2:

Disentangling bacterial invasiveness from lethality in an experimental host-pathogen system
 Biancalani and Jeff Gore
 Mol. Syst. Biol.

Summary

In this work, the authors argue that the commonly used metric for quantifying bacterial virulence, the time taken to kill 50% of infected hosts (LT50), is insufficiently informative of different modes of bacterial virulence. They propose separating virulence into metrics of bacterial lethality and invasiveness, both of which can be extracted from the survival curve. They further separate invasiveness into two regimes, governed by colonization and pathogen growth.

General comments

This is an excellent study. Through well-designed, thorough quantitative analysis, it highlights some limitations of some current metrics in quantifying bacterial pathogenesis. Moreover, the presented phenotypic quantifications provide a starting point for future studies aiming at developing a more mechanistic understanding.

We are happy to read that referee #2 had such a positive reaction to our paper.

One of the most interesting aspects of the work is the constant death rate (of the host), when a bacterial pathogen reaches the carrying capacity. This death rate is largely independent of the bacterial inoculum size. The authors did not attempt to provide a mechanistic explanation for this feature. I understand the advantage in coarse-grained analysis (by modeling and experiment) for the questions that the authors ask. However, I very much hope that the authors would discuss the potential mechanistic basis for their results. For instance, is this behavior a result of quorum sensing? It is well established that QS plays a central role in the pathogenesis of different bacteria, particularly in *P. aeruginosa*. Given their results (particularly the independence of inoculum size), the possibility of QS involvement is rather tempting. Some discussion on this point would be helpful.

Indeed, we observe a death rate independent of the bacterial inoculum size, but did not provide a mechanistic explanation. This is simply because we do not feel that our data provide a definitive explanation. Indeed, *Pseudomonas aeruginosa* is known to be an opportunistic pathogen in human, as it can cause serious lung infections if the patient is (eg) immunocompromised. Our findings suggest that, in the worm, the infection process is memoryless at the measured resolution and within our time scales, which we think is intriguing. It would be enlightening to quantify gene expression of *quorum sensing* proteins during the infection process, but we also feel that this will require significant additional experimental work and would deviate from the scope of the manuscript. In addition, we note that our simple model—which does not include quorum sensing—is sufficient to yield the constant death phase.

From my first impression, their model formulation (on bacterial growth) is somewhat unusual. It appears that it might be more consistent with their experimental protocol by (1) using a simple logistic equation (without "c"), and (2) assuming a positive initial density, corresponding to the inoculum. This question is relevant as it seems that the dissection of the invasion time into colonization time (τ_c) and replication time (τ_r) is tied to the model formulation. By eyeballing, it appears that the experimental data can be equally well fit by using a simple logistic equation, where the initial phase of the growth curve is not bent (Figure 3).

Our first attempts were made using a logistic growth curve and indeed it seems to do an acceptable job in fitting the data (thus, the reviewer is eyeballing well!). The model that we eventually used does a somewhat better job in fitting the data (that's precisely the "bent" phase the reviewer is referring to), and is also more biologically sound which helps with the interpretation of the parameters. In fact, a logistic growth model assumes an initial non-zero amount of pathogen in the

host, which is unphysical because the worms are sterile at the beginning of the experiment. Also, logistic growth assumes that the only growth occurs due to internal replication, but the worms keep feeding on the pathogen lawn for almost the whole duration of the infection process. Therefore, we think that our model with $x(0) = 0$ (sterile worm), and colonization rate c provides a better connection with the biology, but we agree with the referee that the alternate approach is also reasonable.

Other specific comments

1. The paragraph beginning on paragraph 270 seems to muddy the lethality/invasiveness being a property of both host and pathogen, since it establishes that "hosts can cope with pathogens by diminishing their invasiveness (host resistance) or decreasing their lethality (host tolerance)" but then goes on to say that "these two indicators [host resistance and tolerance]... do not account for how fast the pathogen invades," which is odd considering the parallel drawn between reduced invasiveness and increased resistance from the earlier sentence and also makes it seem like invasion speed is a property of pathogen and not host.

Thanks for your comment. Host resistance and host tolerance depend on the pathogen killing rate δ and the host carrying capacity K for that pathogen. The invasion time depends on the host carrying capacity, but also on the pathogen colonization rate c and growth rate r . We therefore feel that the invasion time τ is an indicator that provides complementary information about the infection process. We added the following paragraph in the "Discussion" in the main text:

LINE 290

These two indicators, however, do not account for how fast the pathogen invades, as they depend on K and δ , but not on the pathogen colonization rate c or the pathogen growth rate r . In contrast, our indicator τ provides a timescale for invasion, and might therefore be important for host survival and for the epidemiology of a disease, especially in light of recent findings where pathogen strategies in human disease are disentangled (Regoes *et al*, 2014).

2. Aside from the technical issue associated with the dissection between colonization and replication times, but it's unclear what the utility of splitting the invasion time into these two phases is when it comes to generally understanding the host-pathogen interaction (as opposed to simply noting that colonization and replication rates, not times, affect the invasion time differently). Further elaboration on this point is useful.

On the one hand, the utility of splitting invasion time into colonization and replication times is to validate model predictivity (Fig. 4-C). But this splitting is also important because the colonization rate contains biological information about the host-pathogen interaction. For example, in *C. elegans* mouth, a structure called grinder kills most but not all the pathogenic cells eaten by the worm. The colonization rate of the pathogen is also determined by the pathogen resilience to the grinder which, in turn, depends on cell size, cell wall robustness, etc. We have added the following sentence at the end of the "Results" section:

LINE 254

In addition, the colonization rate c also depends upon the specific host-pathogen pair. For example, it depends on the survival probability of a pathogen cell as it goes through the grinder, the *C. elegans* structure that grind bacterial cells prior to digestion.

3. Nomenclature in the last section is somewhat confusing given that the colonization time is dependent on growth rate, not colonization rate as per equation (4), and Figure 4C shows growth curves with identical "colonization time" but whose invasion times vary with "colonization rate." I wonder whether this is the best approach to these designations but confess I do not have a good alternative since both are meant to indicate a physiological process.

We also have thought about changing our nomenclature, but eventually decided that these phrases were the most accurate description of each phase. Indeed, the model's prediction that colonization time is not a function of the colonization rate is a non-trivial and surprising prediction of the model.

4. Figure S1's caption has a few typos: currently reads "Figures S1" and "the LT50s is". The text box is also too small vertically, cutting off some of the text.

Thanks! We corrected the typos.

5. Line 245 parentheses should maybe read "which follows the invasion phase" instead of "which following"

Thanks again. We corrected the typos.

6. Lines 104-105 make it clear that the metrics reflect both the pathogen and the host; it might be good to add this earlier (line 69-70, where it currently just says pathogen-specific).

We changed into 'host-pathogen specific'.

Reviewer #3:

SUMMARY

The manuscript by Biancalani et al. focuses on modeling and quantifying different steps in bacterial virulence using the nematode *Caenorhabditis elegans* as a system. The authors propose to decompose bacterial virulence in two main phases: invasion and lethality. The authors define "invasion" as the time frame in which the bacterial pathogen colonizes its host and "lethality" (δ) as the time frame where the pathogen kills its host.

We feel that a clarification may be needed here. We do not define the invasion phase as the time frame where the pathogen colonizes and the lethality phase where the pathogen kills the host. This simple and clean description of the phases is only true when the division rate of the bacteria is much larger than the killing rate of the host by the bacteria (at saturation). In fact, our data shows that for some of the pathogens more than 50% of the hosts die during the invasion phase. The end of the invasion phase is determined by the pathogen reaching carrying capacity, which coincides to the entrance of the exponential death phase for the survival function. We edited the main text to improve clarity by adding the following at the beginning of our "Discussion":

LINE 264

The invasion time τ can be thought of as the time taken by the pathogen to reach carrying capacity in the host population. After such time, the host population dies with a constant rate that is given by the lethality δ . For both indicators, we have formulated mathematically precise definitions (see Figure 2 and sub-section B of the result section).

The authors utilize *C. elegans* nematodes exposed to three different bacterial pathogens to develop a model of this process. The authors suggest that bacterial load linked to onset of lethality at specific timepoints that integrate the virulence and host susceptibility. However, the utility of this model is limited in its current form due to minimal effort in its validation, merging of independent host and pathogen variables and limited experimental details to facilitate its reproducibility.

We have expanded the experimental details section in order to facilitate the reproducibility of our results. It should be noted that even though our interdisciplinary approach is unique, the experimental methods are standard. Indeed, there is a massive amount of literature (as discussed in the manuscript) where similar experiments are performed.

Regarding our model: it is indeed a very simple one, because we designed it following the principle of physics which are minimality and simplicity. The role of this model is to elucidate how individual rules give rise to collective behaviors, rather than having a predictive power in experiment (although we welcomed that the model fitted the data in our particular settings). For example, the Ising model of a paramagnet has been extremely useful, yet is extremely simple (atoms are modelled as random variables that only gives ± 1) and not predictive of any paramagnet in nature.

MAJOR POINTS

1) The central premise is gleaned from a dilution experiment of *Pseudomonas*, but the authors fail to acknowledge other confounding factors in the experimental setup.

a. From the host perspective, thin lawns of bacterial food source have been shown to induce caloric restriction and alter immune / stress response pathways, plus could also change pumping and defecation rates that all could influence the rate of "invasion". Were any of these measures examined?

As noted at the beginning of this document, a primary conclusion of our study is that the rate of killing Δ of a species is robust to the dilution of the pathogen. This is surprising precisely for the reasons highlighted by the referee; many things could be changing, yet remarkably the rate of killing in the exponential phase does not change. Therefore, we do not view these as confounding factors, but rather as evidence of why our result is surprising and useful. We quantify the rate of invasion by directly measuring the pathogen load over time. The pathogen load is affected by the defecation/pumping rate and the expression level of immune/stress response pathways (among other things), but the direct measurement of pathogen load is what we feel sufficient to quantify the pathogen load.

b. From the pathogen perspective, virulence factors in all of these pathogens are regulated by quorum sensing and cell density. Thus, each lawn may have dramatically different levels of toxins. Was this measured in any way?

Unfortunately, we did not measure the level of toxins. The toxin level (or other aspect such as the defecation rate) do not confound our claims as they are backed up by direct measurements of pathogen load. We agree that measuring toxin level would be interesting, although it is unfortunately not an easy measure to carry out, and indeed, it is not present in many studies performed with the same system (for example, see seminal paper by Tan et al. 1999). Once again, we highlight that despite the potential changes in quorum sensing (and many other things) in the pathogen, we nonetheless experimentally observe the same rate of killing in the exponential phase.

c. Similarly, It is unclear how the authors link bacterial lawn density to colonization rates and how it affects the model. It is unclear how the authors characterize the different fixed parameters such as the colonization factor c . Was this quantified?

Yes, in Figure 4-C we show how the fixed parameter colonization rate affects the model (solid lines). The figure contains also contains experimental data, which shows how initial pathogen load (mostly due to colonization, since the pathogen has not gotten time to reproduce) changes with the different lawn incubation times.

2) Few variables are included and model lacks true validation set. It is unclear how the model keeps its predictive nature when settings are defined after the experiment. It would have been stronger and more convincing to test the predictivity of the model on a different set of pathogens as the ones used in the present study.

The three pathogens we analyzed have very different virulence mechanisms, and we feel it is a nice result that our simple model agrees so well to data collected on different pathogens. That being said, it would be stronger to test the predictivity of our model to more pathogens which we are planning to do in further studies.

3) As promising as this works seems, the authors only use three bacterial pathogens that have very different virulence mechanisms. Additionally, only one non-pathogen is tested as a negative control, it would have been more informative to test more than the single non-pathogenic bacteria, pathogen virulence mutants (e.g., QS defective) or in more immune-compromised host strains (e.g., *pmk-1*; *glp-4*). Further, in Figure S2, it is confusing why the authors claim that the model does not apply to non-pathogenic bacteria, as it seems that the curves appear to trend toward a similar pattern as the pathogens and

the experiment was ended too soon to make that conclusion as few of the animals have started to die.

We have strengthened our analysis by adding Figure EV4, which shows data from the Fontana lab and the Levine lab.:

Figures EV4: Analysis of experimental data from two other studies suggests that exponential phase is observed in host-pathogen dynamics whereas a more complex scenario arises in senescence. (top) Survival kinetics for the *C. elegans/P. aeruginosa* system, measured using *HandKAchip* – Hands Free Killing Assay on a Chip (data from Erel Levine’s lab, 2016). The survival kinetics exhibits an exponential phase, consistently with our prediction, with exponent $\delta \sim 0.06 h^{-1}$. **(bottom)** Survival kinetics of *C. elegans* wild-type population due to senescence from Walter Fontana’s lab (2013). The survival kinetics, observed for a much larger time range than in our experiments, decay in a non-linear way, which is approximately exponential when restricted to certain time sub-intervals (e.g. two linear fits shown in figure). For age induced-death, the overall death rate of the host population is smaller than those observed in our pathogen-induced mortality experiments.

Regarding the reviewer comment that that our negative control ended “too soon”. As our data show, in our negative control we do not observe an exponential phase for about 9 days (as opposed to the couple of days of the other cases). After such time, worms are quite old and it is hard to push the experiment forward. We therefore felt that we have analyzed the non-pathogenic case for as long as possible (but the data from the Fontana lab does, we believe, help in this regard).

MINOR POINTS

4) Many of the details surrounding the central experiments could be better delineated. For example, the actual concentrations of bacteria used in any of the experiments is not noted, nor is it clear whether a comparable amount is used for each of the pathogens.

We closely follow the culture procedure described in the method papers that we cited. For increasing clarity, we have provided additional information of our experimental details. We have added the following sentence:

LINE 384

Pathogen monocultures are grown for 24 hr at 30°C in 5mL LB, after which the monocultures reach saturation.

More changes to the experimental section are outlined below.

5) Microscopy-based (FISH or otherwise) visualization of bacterial loads would help ensure that the overall levels are reflective of the population and not due to a jackpotting effect by a few highly colonized animals.

In a previous paper (Vega and Gore, PLOS Biology (2017)) we have explored some of the consequences of stochastic colonization and heterogeneity between worms. We are therefore aware that there can be significant heterogeneity in colonization levels, but once again we believe that this is another example of why it is surprising that such a simple model seems to work to describe the survival of worms exposed to different pathogens. In addition, we note that microscopy based measures of bacterial loads can be misleading due to the counting of dead cells:

<https://link.springer.com/article/10.1007/s12275-013-2589-8>

6) Interpretation is limited by the fact that few / no bona fide (reagent) controls are included in the results. Were the antibiotics effective? How quickly were animals paralyzed at the different stages? Also, for one of the central assays (colonization), premixing of the 30 animals seems likely to normalize potential animal-animal variation that may bias interpretation. Lastly, it is not clear how the precise numbers of animals were transferred, whether the animal number was counted or rather inferred.

We have modified the experimental section of the paper to add more details. We have added a supplementary table containing the IDs of the reagents we use:

Reagent/Resource	Reference or Source	Identifier or Catalog Number
Experimental Models		
PA14 (P. aeruginosa)	Ausubel's lab (Harvard)	
Db10 (S. marcescens)	CGC	
LT2 (S. enterica)	CGC	
OP50 (E. coli)	CGC	
SS104 (C. elegans)	CGC	glp-4(bn2)
Chemicals, Enzymes and other reagents (e.g. drugs, peptides, recombinant proteins, dyes etc.)		
Triton X-100	Sigma-Aldrich	X100-5ML
Levamisole hydrochloride	Sigma-Aldrich	#1359302
Gentamicin sulfate	Sigma-Aldrich	#1289003
Carbenicillin disodium salt	Sigma-Aldrich	C1389
Poly(ethylene glycol) methyl ether	Sigma-Aldrich	#81316
Software		
Wolfram Mathematica 11	https://www.wolfram.com/mathematica/	
R 3.5.2	https://www.r-project.org/	
Other		
Kimble Kontes tubes	Grainger	# 6HAY2
Pellet pestles	Sigma-Aldrich	Z359947
Pellet pestles Cordless motor	Sigma-Aldrich	Z359971

2

To respond to the reviewer concerns:

- Antibiotics were indeed effective.
- With our concentration of levamisole the animals are paralyzed in less than a second.
- The 30 animals were counted and transferred using a pipette + M9PG. Normalizing animal-animal variation is intentional and in our opinion does not hinder interpretability.

We have expanded two sentences in the “Pathogen load assay” section, which now read:

To measure the pathogen load, we collected each day 30 alive worms by washing the plates with M9WB. We then washed worm resuspension them four times in TXLV. Due to levamisole, worm peristalsis is interrupted in a few seconds, and the mouth and anus of the worm remains shut, thus preventing the internal bacteria to be flushed out.

7) Similarly, non-normalized values for colonization levels should also be presented to ensure values are within the realm of published experiments.

They were already present in the supplementary tables, therefore we felt that no change was necessary for this point.

8) A discussion of the relevance of these results to natural pathogens and commensals of *C. elegans* would be helpful for applicability.

We have added the following sentence to our discussion:

LINE 339

Recent works have demonstrated that *C. elegans* is a suitable model system for microbiome investigations (Zhang *et al*, 2017)(Vega & Gore, 2017), and demonstrated the profound effects that gut commensal bacteria might have in fending off pathogen of affecting host life span (Clark & Hodgkin, 2014).

And also modified the conclusion which now reads:

LINE 347

In future work, it would be interesting to pre-colonize the host population with a commensal bacteria, before exposing the population to a pathogen. In this way, it would be possible to investigate how the invasion time τ and the lethality δ change in the pre-colonized population, thus revealing in a quantitative fashion the role of the microbiome in fending off the pathogen. It is our hope that simple quantitative laws of host-pathogen dynamics revealed by our system may provide insight into pathogenesis in more complex host-pathogen systems.

2nd Editorial Decision

2nd May 2019

Thank you for sending us your revised manuscript. We have now heard back from the three referees who were asked to evaluate your study. As you will see below, the reviewers think that the study has improved as a result of the performed revisions. However, reviewer #3 raises a couple of remaining issues, which we would ask you to address in a minor revision.

REFeree REPORTS.

Reviewer #1:

Referee report on the second version of Biancalani and Gore manuscript

I carefully read all reviewer comments, the authors reply and the (minimally) modified manuscript.

I had only few comments and it seems that the authors have provided adequate (though minimalistic) answers to all of them. Their comment that the exponential death phase may be fairly specific to *C. elegans* (or to other organisms with no adaptive immunity) may limit the generality of the work, but I feel it is still a good work that shows how a careful dissection of population dynamics uncovers the more biologically relevant parameters.

Beside the conceptual clarification of pathogen induced mortality, the practical utility of the division between invasion and lethality stages would depend on follow-up works that would study the separate impact of different host-pathogen mechanisms on invasion and lethality. I hope that this, theoretically oriented, work will inspire others to perform such experiments.

I approve the manuscript's publication

Reviewer #2:

The authors have fully addressed my raised issues and I support the publication of the work at the journal.

Reviewer #3:

The authors have addressed each point brought up by the reviewers and have appropriately answered most of them. We welcome the authors' efforts in providing more experimental details to facilitate reproducibility. There are still a few minor comments on the following points:

- 1) By adding the figure EV4 showing data from the Levine et al., 2016 and the Fontana et al., 2013 studies the authors provide supplementary support showing survival kinetics difference between *C. elegans* survival rates when exposed or not to pathogens. The data raised by the Fontana study raise one question. The study unique experimental setting generated high-resolution data both by the number of time points used and the number of individual nematodes screened. This resolution raises the question if the data used by the authors was enough to resolve a similar pattern. It would be interesting to redo the analysis using the lifespan machine data (or more densely collected data).
- 2) Additionally, the Fontana study provides a good data set to explore the effect of non-pathogenic bacteria on the host natural death rate. The *E. coli* OP50 as certainly the less effect on its host, and as the authors suggested and did on their manuscript it would be interesting to try to further expose *C. elegans* to a wider range of non-pathogenic bacteria to confirm that the pattern holds.
- 3) The authors mention that antibiotics were effective without providing any more details.

2nd Revision - authors' response

15th May 2019

RESPONSE TO REVIEWER 3

The authors have addressed each point brought up by the reviewers and have appropriately answered most of them. We welcome the authors' efforts in providing more experimental details to facilitate reproducibility. There are still a few minor comments on the following points:

- 1) By adding the figure EV4 showing data from the Levine et al., 2016 and the Fontana et al., 2013 studies the authors provide supplementary support showing survival kinetics difference between *C. elegans* survival rates when exposed or not to pathogens. The data raised by the Fontana study raise one question. The study unique experimental setting generated high-resolution data both by the number of time points used and the number of individual nematodes screened. This resolution raises the question if the data used by the authors was enough to resolve a similar pattern. It would be interesting to redo the analysis using the lifespan machine data (or more densely collected data).

We agree that it would be interesting to repeat the experiment using the lifespan machine, as increased temporal resolution and number of worms open the possibility for discriminating additional complexities within the data. However, moving to an entirely different experimental setup, with all of the associated experimental issues that would likely arise, would likely be an entirely new project. We believe that the data that we have presented is sufficient to justify our primary claim, which is that our data shows an exponential phase in the survival curves. It is indeed possible that the survivability decay may show a weak non-linear regime with additional resolution, yet its linear approximation (*i.e.* the coefficient δ) remains the best estimate for pathogen lethality. Therefore, we feel that the claims in our manuscript are substantiated by our data in their present form.

- 2) Additionally, the Fontana study provides a good data set to explore the effect of non-pathogenic bacteria on the host natural death rate. The *E. coli* OP50 as certainly the less effect on its host, and as the authors suggested and did on their manuscript it would be interesting to try to further expose

C. elegans to a wider range of non-pathogenic bacteria to confirm that the pattern holds.

We agree as well that this is a most interesting experiments we would like to have in a follow-up study. Given the timeline requested for resubmission, it is not possible to pursue this here.

3) The authors mention that antibiotics were effective without providing any more details.

Thank you for pointing this out to us – we missed this minor point in previous response. We edited main text as follows:

LINE 381: To test for antibiotic efficacy, we grind the worm population (see *Pathogen load assay* subsection) and plate the solution on three LB agar plates which we then incubate at 30 C for 48 hours. Antibiotic is considered effective as we did not find any colony forming units in agar plates.

Accepted

17th May 2019

Thank you again for sending us your revised manuscript. We are now satisfied with the modifications made and I am pleased to inform you that your paper has been accepted for publication.

Corresponding Author Name: Jeff Gore

Journal Submitted to: Molecular System Biology

Manuscript Number: MSB-18-8707